# Combinatorial Bayesian Optimization using the Graph Cartesian Product

**Changyong Oh[1] Jakub M. Tomczak[2] Efstratios Gavves[1] Max Welling[1,2,3]**
[1] University of Amsterdam [2] Qualcomm AI Research [3] CIFAR
C.Oh@uva.nl, jtomczak@qti.qualcomm.com, egavves@uva.nl, m.welling@uva.nl

## Abstract

This paper focuses on Bayesian Optimization (BO) for objectives on combinatorial search spaces, including ordinal and categorical variables. Despite the abundance of potential applications of Combinatorial BO, including chipset configuration search and neural architecture search, only a handful of methods have been proposed. We introduce COMBO, a new Gaussian Process (GP) BO. COMBO quantifies "smoothness" of functions on combinatorial search spaces by utilizing a *combinatorial graph*. The vertex set of the combinatorial graph consists of all possible joint assignments of the variables, while edges are constructed using the graph Cartesian product of the sub-graphs that represent the individual variables. On this combinatorial graph, we propose an ARD diffusion kernel with which the GP is able to model high-order interactions between variables leading to better performance. Moreover, using the Horseshoe prior for the scale parameter in the ARD diffusion kernel results in an effective variable selection procedure, making COMBO suitable for high dimensional problems. Computationally, in COMBO the graph Cartesian product allows the Graph Fourier Transform calculation to scale linearly instead of exponentially.We validate COMBO in a wide array of realistic benchmarks, including weighted maximum satisfiability problems and neural architecture search. COMBO outperforms consistently the latest state-of-the-art while maintaining computational and statistical efficiency.

## 1 Introduction

This paper focuses on Bayesian Optimization (BO) [42] for objectives on combinatorial search spaces consisting of ordinal or categorical variables. Combinatorial BO [21] has many applications, including finding optimal chipset configurations, discovering the optimal architecture of a deep neural network or the optimization of compilers to embed software on hardware optimally. All these applications, where Combinatorial BO is potentially useful, feature the following properties. They *(i)* have black-box objectives for which gradient-based optimizers [47] are not amenable, *(ii)* have expensive evaluation procedures for which methods with low sample efficiency such as, evolution [12] or genetic [9] algorithms are unsuitable, and *(iii)* have noisy evaluations and highly non-linear objectives for which simple and exact solutions are inaccurate [5, 11, 40].

Interestingly, most BO methods in the literature have focused on continuous [29] rather than combinatorial search spaces. One of the reasons is that the most successful BO methods are built on top of Gaussian Processes (GPs) [22, 33, 42]. As GPs rely on the smoothness defined by a kernel to model uncertainty [37], they are originally proposed for, and mostly used in, continuous input spaces. In spite of the presence of kernels proposed on combinatorial structures [17, 25, 41], to date the relation between the smoothness of graph signals and the smoothness of functions defined on combinatorial structures has been overlooked and not been exploited for BO on combinatorial structures. A simple solution is to use continuous kernels and round them up. This rounding, however, is not incorporated

when computing the covariances at the next BO iteration [14], leading to unwanted artifacts. Furthermore, when considering combinatorial search spaces the number of possible configurations quickly explodes: for $M$ categorical variables with $k$ categories the number of possible combinations scales with $\mathcal{O}(k^M)$. Applying BO with GPs on combinatorial spaces is, therefore, not straightforward.

We propose COMBO, a novel Combinatorial BO designed to tackle the aforementioned problems of lack of smoothness and computational complexity on combinatorial structures. To introduce smoothness of a function on combinatorial structures, we propose the *combinatorial graph*. The combinatorial graph comprises sub-graphs –one per categorical (or ordinal) variable– later combined by the graph Cartesian product. The combinatorial graph contains as vertices all possible combinatorial choices. We define then smoothness of functions on combinatorial structures to be the smoothness of graph signals using the Graph Fourier Transform (GFT) [35]. Specifically, we propose as our GP kernel on the graph a variant of the diffusion kernel, the automatic relevance determination(ARD) diffusion kernel, for which computing the GFT is computationally tractable via a decomposition of the eigensystem. With a GP on a graph COMBO accounts for arbitrarily high order interactions between variables. Moreover, using the sparsity-inducing Horseshoe prior [6] on the ARD parameters COMBO performs variable selection and scales up to high-dimensional. COMBO allows for accurate, efficient and large-scale BO on combinatorial search spaces.

In this work, we make the following contributions. First, we show how to introduce smoothness on combinatorial search spaces by introducing combinatorial graphs. On top of a combinatorial graph we define a kernel using the GFT. Second, we present an algorithm for Combinatorial BO that is computationally scalable to high dimensional problems. Third, we introduce individual scale parameters for each variable making the diffusion kernel more flexible. When adopting a sparsity inducing Horseshoe prior [6, 7], COMBO performs variable selection which makes it scalable to high dimensional problems. We validate COMBO extensively on *(i)* four numerical benchmarks, as well as two realistic test cases: *(ii)* the weighted maximum satisfiability problem [16, 39], where one must find boolean values that maximize the combined weights of satisfied clauses, that can be made true by turning on and off the variables in the formula, *(iii)* neural architecture search [10, 48]. Results show that COMBO consistently outperforms all competitors.

## 2 Method

### 2.1 Bayesian optimization with Gaussian processes

Bayesian optimization (BO) aims at finding the global optimum of a black-box function $f$ over a search space $\mathcal{X}$, namely, $\mathbf{x}_{opt} = \arg\min_{\mathbf{x} \in \mathcal{X}} f(\mathbf{x})$. At each round, a surrogate model attempts to approximate $f(\mathbf{x})$ based on the evaluations so far, $\mathcal{D} = \{(\mathbf{x}_i, y_i = f(\mathbf{x}_i))\}$. Then an acquisition function suggests the most promising point $\mathbf{x}_{i+1}$ that should be evaluated. The $\mathcal{D}$ is appended by the new evaluation, $\mathcal{D} = \mathcal{D} \cup (\{\mathbf{x}_{i+1}, y_{i+1}\})$. The process repeats until the evaluation budget is depleted.

The crucial design choice in BO is the surrogate model that models $f(\cdot)$ in terms of *(i)* a predictive mean to predict $f(\cdot)$, and *(ii)* a predictive variance to quantify the prediction uncertainty. With a GP surrogate model, we have the predictive mean $\mu(\mathbf{x}_* \,|\, \mathcal{D}) = K_{* \mathcal{D}} (K_{\mathcal{D}\mathcal{D}} + \sigma_n^2 I)^{-1} \mathbf{y}$ and variance $\sigma^2(\mathbf{x}_* \,|\, \mathcal{D}) = K_{**} - K_{* \mathcal{D}} (K_{\mathcal{D}\mathcal{D}} + \sigma_n^2 I)^{-1} K_{\mathcal{D} *}$ where $K_{**} = K(\mathbf{x}_*, \mathbf{x}_*)$, $[K_{* \mathcal{D}}]_{1,i} = K(\mathbf{x}_*, \mathbf{x}_i)$, $K_{\mathcal{D} *} = (K_{* \mathcal{D}})^T$, $[K_{\mathcal{D}\mathcal{D}}]_{i,j} = K(\mathbf{x}_i, \mathbf{x}_j)$ and $\sigma_n^2$ is the noise variance.

### 2.2 Combinatorial graphs and kernels

In BO on continuous search spaces the most popular surrogate models rely on GPs [22, 33, 42]. Their popularity does not extend to combinatorial spaces, although kernels on combinatorial structures have also been proposed [17, 25, 41]. To design an effective GP-based BO algorithm on combinatorial structures, a space of smooth functions –defined by the GP– is needed. We circumvent this requirement by the notion of the combinatorial graph defined as a graph, which contains all possible combinatorial choices as its vertices for a given combinatorial problem. That is, each vertex corresponds to a different joint assignment of categorical or ordinal variables. If two vertices are connected by an edge, then their respective set of combinatorial choices differ only by a single combinatorial choice. As a consequence, we can now revisit the notion of smoothness on combinatorial structures as smoothness of a graph signal [8, 35] defined on the combinatorial graph. On a combinatorial graph, the shortest path is closely related to the Hamming distance.

**The combinatorial graph** To construct the combinatorial graph, we first define one sub-graph per combinatorial variable $C_i$, $\mathcal{G}(C_i)$. For a categorical variable $C_i$, the sub-graph $\mathcal{G}(C_i)$ is chosen to be a complete graph while for an ordinal variable we have a path graph. We aim at building a search space for combinatorial choices, *i.e.*, a combinatorial graph, by combining sub-graphs $G(C_i)$ in such way that a distance between two adjacent vertices corresponds to a change of a value of a single combinatorial variable. It turns out that the graph Cartesian product [15] ensures this property. Then, the graph Cartesian product of subgraphs $\mathcal{G}(C_j) = (\mathcal{V}_j, \mathcal{E}_j)$ is defined as $\mathcal{G} = (\mathcal{V}, \mathcal{E}) = \square_i \mathcal{G}(C_i)$, where $\mathcal{V} = \times_i \mathcal{V}_i$ and $(v_1 = (c_1^{(1)}, \cdots, c_N^{(1)}), v_2 = (c_1^{(2)}, \cdots, c_N^{(2)})) \in \mathcal{E}$ if and only if $\exists j$ such that $\forall i \neq j \ c_i^{(1)} = c_i^{(2)}$ and $(c_j^{(1)}, c_j^{(2)}) \in \mathcal{E}_j$.

As an example, let us consider a simplistic hyperparameter optimization problem for learning a neural network with three combinatorial variables: *(i)* the batch size, $c_1 \in C_1 = \{16, 32, 64\}$, *(ii)* the optimizer $c_2 \in C_2 = \{AdaDelta, RMSProp, Adam\}$ and *(iii)* the learning rate annealing $c_3 \in C_3 = \{Constant, Annealing\}$. The sub-graphs $\{\mathcal{G}(C_i)\}_{i=1,2,3}$ for each of the combinatorial variables, as well as the final combinatorial graph after the graph Cartesian product, are illustrated in Figure 1. For the ordinal batch size variable we have a path graph, whereas for the categorical optimizer and learning rate annealing variables we have complete graphs. The final combinatorial graph contains all possible combinations for batch size, optimizer and learning rate annealing.

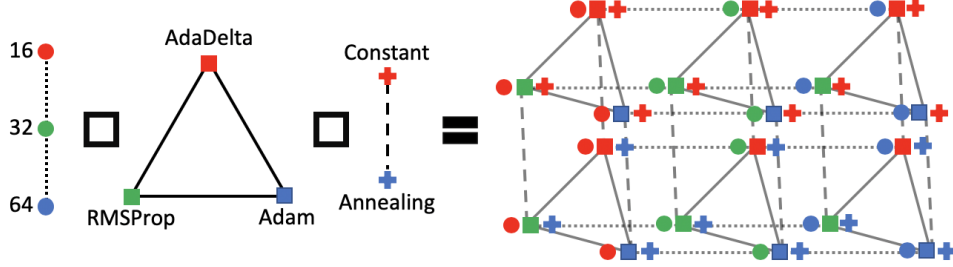

Figure 1: Combinatorial Graph: graph Cartesian product of sub-graphs $\mathcal{G}(C_1) \square \mathcal{G}(C_2) \square \mathcal{G}(C_3)$

**Cartesian product and Hamming distance** The Hamming distance is a natural choice of distance on categorical variables. With all complete sub-graphs, the shortest path between two vertices in the combinatorial graph is exactly equivalent to the Hamming distance between the respective categorical choices.

**Theorem 2.2.1.** *Assume a combinatorial graph $\mathcal{G} = (\mathcal{V}, \mathcal{E})$ constructed from categorical variables, $C_1, \ldots, C_N$, that is, $\mathcal{G}$ is a graph Cartesian product $\square_i \mathcal{G}(C_i)$ of complete sub-graphs $\{\mathcal{G}(C_i)\}_i$. Then the shortest path $s(v_1, v_2; \mathcal{G})$ between vertices $v_1 = (c_1^{(1)}, \cdots, c_N^{(1)}), v_2 = (c_1^{(2)}, \cdots, c_N^{(2)}) \in \mathcal{V}$ on $\mathcal{G}$ is equal to the Hamming distance between $(c_1^{(1)}, \cdots, c_N^{(1)})$ and $(c_1^{(2)}, \cdots, c_N^{(2)})$.*

*Proof.* The proof of Theorem 2.2.1 could be found in Supp. 1 $\qquad\qquad\qquad\square$

When there is a sub-graph which is not complete, the below result follows from the Thm. 2.2.1:

**Corollary 2.2.1.** *If a sub-graph is not a complete graph, then the shortest path is equal to or bigger than the Hamming distance.*

The combinatorial graph using the graph Cartesian product is a natural search space for combinatorial variables that can encode a widely used metric on combinatorial variables like Hamming distance.

**Kernels on combinatorial graphs.** In order to define the GP surrogate model for a combinatorial problem, we need to specify a a proper kernel on a combinatorial graph $\mathcal{G} = (\mathcal{V}, \mathcal{E})$. The role of the surrogate model is to smoothly interpolate and extrapolate neighboring data. To define a smooth function on a graph, *i.e.*, a smooth graph signal $f : \mathcal{V} \mapsto \mathbb{R}$, we adopt Graph Fourier Transforms (GFT) from graph signal processing [35]. Similar to Fourier analysis on Euclidean spaces, GFT can represent any graph signal as a linear combination of graph Fourier bases. Suppressing the high frequency modes of the eigendecomposition approximates the signal with a smooth function on the graph. We adopt the diffusion kernel which penalizes basis-functions in accordance with the magnitude of the frequency [25, 41].

To compute the diffusion kernel on the combinatorial graph $\mathcal{G}$, we need the eigensystem of graph Laplacian $L(\mathcal{G}) = \mathbf{D}_{\mathcal{G}} - \mathbf{A}_{\mathcal{G}}$, where $\mathbf{A}_{\mathcal{G}}$ is the adjacency matrix and $\mathbf{D}_{\mathcal{G}}$ is the degree matrix of the graph $\mathcal{G}$. The eigenvalues $\{\lambda_1, \lambda_2, \cdots, \lambda_{|\mathcal{V}|}\}$ and eigenvectors $\{u_1, u_2, \cdots, u_{|\mathcal{V}|}\}$ of the graph

Laplacian $L(\mathcal{G})$ are the graph Fourier frequencies and bases, respectively. Eigenvectors paired with large eigenvalues correspond to high-frequency Fourier bases. The diffusion kernel is defined as

$$k([p],[q]|\beta) = \sum_{i=1}^{n} e^{-\beta\lambda_i} u_i([p])u_i([q]), \tag{1}$$

from which it is clear that higher frequencies, $\lambda_i \gg 1$, are penalized more. In a matrix form, with $\Lambda = diag(\lambda_1, \cdots, \lambda_{|\mathcal{V}|})$ and $\mathbf{U} = [u_1, \cdots, u_{|\mathcal{V}|}]$, the kernel takes the following form:

$$K(\mathcal{V},\mathcal{V}) = \mathbf{U}\exp(-\beta\Lambda)\mathbf{U}^T, \tag{2}$$

which is the Gram matrix on all vertices whose submatrix is the Gram matrix for a subset of vertices.

## 2.3  Scalable combinatorial Bayesian optimization with the graph Cartesian product

The direct computation of the diffusion kernel is infeasible because it involves the eigendecomposition of the Laplacian $L(\mathcal{G})$, an operation with cubic complexity with respect to the number of vertices $|\mathcal{V}|$. As we rely on the graph Cartesian product $\square_i \mathcal{G}_i$ to construct our combinatorial graph, we can take advantage of its properties and dramatically increase the efficiency of the eigendecomposition of the Laplacian $L(\mathcal{G})$. Further, due to the construction of the combinatorial graph, we can propose a variant of the diffusion kernel: automatic relevance determination (ARD) diffusion kernel. The ARD diffusion kernel has more flexibility in its modeling capacity. Moreover, in combination with the sparsity-inducing Horseshoe prior [6] the ARD diffusion kernel performs variable selection automatically that allows to scale to high dimensional problems.

**Speeding up the eigendecomposition with graph Cartesian products.** Direct computation of the eigensystem of the Laplacian $L(\mathcal{G})$ naively is infeasible, even for problems of moderate size. For instance, for 15 binary variables, eigendecomposition complexity is $O(|\mathcal{V}|^3) = (2^{15})^3$.

The graph Cartesian product allows to improve the scalability of the eigendecomposition. The Laplacian of the Cartesian product of two sub-graphs $\mathcal{G}_1$ and $\mathcal{G}_2$, $\mathcal{G}_1\square\mathcal{G}_2$, can be algebraically expressed using the Kronecker product $\otimes$ and the Kronecker sum $\oplus$ [15]:

$$L(\mathcal{G}_1 \square \mathcal{G}_2) = L(\mathcal{G}_1) \oplus L(\mathcal{G}_2) = L(\mathcal{G}_1) \otimes \mathbf{I}_1 + \mathbf{I}_2 \otimes L(\mathcal{G}_2), \tag{3}$$

where $\mathbf{I}$ denotes the identity matrix. Considering the eigensystems $\{(\lambda_i^{(1)}, u_i^{(1)})\}$ and $\{(\lambda_j^{(2)}, u_j^{(2)})\}$ of $\mathcal{G}_1$ and $\mathcal{G}_2$, respectively, the eigensystem of $\mathcal{G}_1\square\mathcal{G}_2$ is $\{(\lambda_i^{(1)} + \lambda_j^{(2)}, u_i^{(1)} \otimes u_j^{(2)})\}$. Given Eq. (3) and matrix exponentiation, for the diffusion kernel of $m$ categorical (or ordinal) variables we have

$$\mathbf{K} = \exp\left(-\beta \bigoplus_{i=1}^{m} L(\mathcal{G}_i)\right) = \bigotimes_{i=1}^{m} \exp\left(-\beta L(\mathcal{G}_i)\right). \tag{4}$$

This means we can compute the kernel matrix by calculating the Kronecker product per sub-graph kernel. Specifically, we obtain the kernel for the $i$-th sub-graph from the eigendecomposition of its Laplacian as per eq. (2).

Importantly, the decomposition of the final kernel into the Kronecker product of individual kernels in Eq. (4) leads to the following proposition.

**Proposition 2.3.1.** *Assume a graph $\mathcal{G} = (\mathcal{V},\mathcal{E})$ is the graph cartesian product of sub-graphs $\mathcal{G} = \square_i,\mathcal{G}_i$. The graph Fourier Transform of $\mathcal{G}$ can be computed in $O(\sum_{i=1}^{m} |\mathcal{V}_i|^3)$ while the direct computation takes $O(\prod_{i=1}^{m} |\mathcal{V}_i|^3)$.*

*Proof.* The proof of Proposition 2.3.1 could be found in the Supp. 1.

**Variable-wise edge scaling.** We can make the kernel more flexible by considering individual scaling factors $\{\beta_i\}$, a single $\beta_i$ for each variable. The diffusion kernel then becomes:

$$\mathbf{K} = \exp\left(\bigoplus_{i=1}^{m} -\beta_i L(\mathcal{G}_i)\right) = \bigotimes_{i=1}^{m} \exp\left(-\beta_i L(\mathcal{G}_i)\right), \tag{5}$$

where $\beta_i \geq 0$ for $i = 1, \ldots, m$. Since the diffusion kernel is a discrete version of the exponential kernel, the application of the individual $\beta_i$ for each variable is equivalent to the ARD kernel [27, 31]. Hence, we can perform variable (sub-graph) selection automatically. We refer to this kernel as the *ARD diffusion kernel*.

**Prior on $\beta_i$.** To determine $\beta_i$, and to prevent GP with ARD kernel from overfitting, we apply posterior sampling with a Horseshoe prior [6] on the $\{\beta_i\}$. The Horseshoe prior encourages sparsity, and, thus, enables variable selection, which, in turn, makes COMBO statistically scalable to high dimensional problems. For instance, if $\beta_i$ is set to zero, then $L(\mathcal{G}_i)$ does not contribute in Eq (5).

---

**Algorithm 1** COMBO: Combinatorial Bayesian Optimization on the combinatorial graph

---
 1: **Input:** $N$ **combinatorial variables** $\{C_i\}_{i=1,\cdots,N}$
 2: Set a search space and compute Fourier frequencies and bases: # See Sect. 2.2
 3: ▷ Set sub-graphs $\mathcal{G}(C_i)$ for each variables $C_i$.
 4: ▷ Compute eigensystem $\{(\lambda_k^{(i)}, u_k^{(i)})\}_{i,k}$ for each sub-graph $\mathcal{G}(C_i)$
 5: ▷ Construct the combinatorial graph $\mathcal{G} = (\mathcal{V}, \mathcal{E}) = \square_i \mathcal{G}(C_i)$ using graph Cartesian product.
 6: Initialize $\mathcal{D}$.
 7: **repeat**
 8:   Fit GP using ARD diffusion kernel to $\mathcal{D}$ with slice sampling : $\mu(v_* | \mathcal{D}), \sigma^2(v_* | \mathcal{D})$
 9:   Maximize acquisition function : $v_{next} = \arg \max_{v_* \in \mathcal{V}} a(\mu(v_* | \mathcal{D}), \sigma^2(v_* | \mathcal{D}))$
10:   Evaluate $f(v_{next})$, append to $\mathcal{D} = \mathcal{D} \cup \{(v_{next}, f(v_{next}))\}$
11: **until** stopping criterion

---

### 2.4 COMBO **algorithm**

We present the COMBO approach in Algorithm 1. More details about COMBO could be found in the Supp. Sections 2 and 3.

We start the algorithm with defining all sub-graphs. Then, we calculate GFT (line 4 of Alg. 1), whose result is needed to compute the ARD diffusion kernel, which could be sped up due to the application of the graph Cartesian product. Next, we fit the surrogate model parameters using slice sampling [30, 32] (line 8). Sampling begins with 100 steps of the burn-in phase. With the updated $\mathcal{D}$ of evaluated data, 10 points are sampled without thinning. More details on the surrogate model fitting are given in Supp. 2.

Last, we maximize the acquisition function to find the next point for evaluation (line 9). For this purpose, we begin with evaluating 20,000 randomly selected vertices. Twenty vertices with highest acquisition values are used as initial points for acquisition function optimization. We use the breadth-first local search (BFLS), where at a given vertex we compare acquisition values on adjacent vertices. We then move to the vertex with the highest acquisition value and repeat until no acquisition value on adjacent vertices are higher than the acquisition value at the current vertex. BFLS is a local search, however, the initial random search and multi-starts help to escape from local minima. In experiments (Supp. 3.1) we found that BFLS performs on par or better than non-local search, while being more efficient.

In our framework we can use any acquisition function like GP-UBC, the Expected Improvement (EI) [37], the predictive entropy search [18] or knowledge gradient [49]. We opt for EI that generally works well in practice [40].

## 3 Related work

While for continuous inputs, $\mathcal{X} \subseteq \mathbb{R}^D$, there exist efficient algorithms to cope with high-dimensional search spaces using Gaussian processes(GPs) [33] or neural networks [44], few Bayesian Optimization(BO) algorithms have been proposed for combinatorial search spaces [2, 4, 20].

A basic BO approach to combinatorial inputs is to represent all combinatorial variables using one-hot encoding and treating all integer-valued variables as values on a real line. Further, for the integer-valued variables an acquisition function considers the closest integer for the chosen real value. This approach is used in Spearmint [42]. However, applying this method naively may result in severe problems, namely, the acquisition function could repeatedly evaluate the same points due to rounding real values to an integer and the one-hot representation of categorical variables. As pointed out in [14], this issue could be fixed by making the objective constant over regions of input variables for which the actual objective has to be evaluated. The method was presented on a synthetic problem with two integer-valued variables, and a problem with one categorical variable and one integer-valued variable. Unfortunately, it remains unclear whether this approach is suitable for high-dimensional problems. Additionally, the proposed transformation of the covariance function seems to be better suited for ordinal-valued variables rather than categorical variables, further restricting the utility of this approach. In contrast, we propose a method that can deal with high-dimensional combinatorial (categorical and/or ordinal) spaces.

Another approach to combinatorial optimization was proposed in BOCS [2] where the sparse Bayesian linear regression was used instead of GPs. The acquisition function was optimized by a semi-definite programming or simulated annealing that allowed to speed up the procedure of picking new points for next evaluations. However, BOCS has certain limitations which restrict its application mostly to problems with low order interactions between variables. BOCS requires users to specify the highest order of interactions among categorical variables, which inevitably ignores interaction terms of orders higher than the user-specified order. Moreover, due to its parametric nature, the surrogate model of BOCS has excessively large number of parameters even for moderately high order (*e.g.*, up to the 4th or 5th order). Nevertheless, this approach achieved state-of-the-art results on four high-dimensional binary optimization problems. Different from [2], we use a non-parametric regression, *i.e.*, GPs and perform variable selection both of which give more statistical efficiency.

## 4 Experiments

We evaluate COMBO on two binary variable benchmarks, one ordinal and one multi-categorical variable benchmarks, as well as in two realistic problems: weighted Maximum Satisfiability and Neural Architecture Search. We convert all into minimization problems. We compare SMAC [20], TPE [4], Simulated Annealing (SA) [45], as well as with BOCS (BOCS-SDP and BOCS-SA3)[1] [2]. All details regarding experiments, baselines and results are in the supplementary material. The code is available at: `https://github.com/QUVA-Lab/COMBO`

### 4.1 Bayesian optimization with binary variables [2]

Table 1: Results on the binary benchmarks (Mean $\pm$ Std.Err. over 25 runs)

| METHOD | CONTAMINATION CONTROL | | | ISING SPARSIFICATION | | |
|---|---|---|---|---|---|---|
| | $\lambda = 0$ | $\lambda = 10^{-4}$ | $\lambda = 10^{-2}$ | $\lambda = 0$ | $\lambda = 10^{-4}$ | $\lambda = 10^{-2}$ |
| SMAC | 21.61$\pm$0.04 | 21.50$\pm$0.03 | 21.68$\pm$0.04 | 0.152$\pm$0.040 | 0.219$\pm$0.052 | 0.350$\pm$0.045 |
| TPE | 21.64$\pm$0.04 | 21.69$\pm$0.04 | 21.84$\pm$0.04 | 0.404$\pm$0.109 | 0.444$\pm$0.095 | 0.609$\pm$0.107 |
| SA | 21.47$\pm$0.04 | 21.49$\pm$0.04 | 21.61$\pm$0.04 | **0.095**$\pm$0.033 | 0.117$\pm$0.035 | 0.334$\pm$0.064 |
| BOCS-SDP | 21.37$\pm$0.03 | 21.38$\pm$0.03 | 21.52$\pm$0.03 | 0.105$\pm$0.031 | **0.059**$\pm$0.013 | **0.300**$\pm$0.039 |
| COMBO | **21.28**$\pm$0.03 | **21.28**$\pm$0.03 | **21.44**$\pm$0.03 | 0.103$\pm$0.035 | 0.081$\pm$0.028 | 0.317$\pm$0.042 |

**Contamination control**    The contamination control in food supply chain is a binary optimization problem with 21 binary variables ($\approx 2.10 \times 10^6$ configurations) [19], where one can intervene at each stage of the supply chain to quarantine uncontaminated food with a cost. The goal is to minimize food contamination while minimizing the prevention cost. We set the budget to 270 evaluations including 20 random initial points. We report results in Table 1 and figures in Supp. 4.1.2. COMBO outperforms all competing methods. Although the optimizing variables are binary, there exist higher order interactions among the variables due to the sequential nature of the problem, showcasing the importance of the modelling flexibility of COMBO.

**Ising sparsification**    A probability mass function(p.m.f) $p(x)$ can be defined by an Ising model $I_p$. In Ising sparsification, we approximate the p.m.f $p(z)$ of $I_p$ with a p.m.f $q(z)$ of $I_q$. The objective is the KL-divergence between $p$ and $q$ with a $\lambda$-parameterized regularizer: $\mathcal{L}(x) = D_{KL}(p||q) + \lambda \|x\|_1$. We consider 24 binary variable Ising models on $4 \times 4$ spin grid ($\approx 1.68 \times 10^7$ configurations) with a budget of 170 evaluations, including 20 random initial points. We report results in Table 1 and figures in Supp. 4.1.1. We observe that COMBO is competitive, obtaining slightly worse results, probably because in Ising sparsification there exist no complex interactions between variables.

## 4.2 Bayesian optimization with ordinal and multi-categorical variables

**Ordinal variables** The Branin benchmark is an optimization problem of a non-linear function over a 2D search space [21]. We discretize the search space, namely, we consider a grid of points that leads to an optimization problem with ordinal variables. We set the budget to 100 evaluations and report results in Table 2 and Figure 9 in the Supp. COMBO converges to a better solution faster and with better stability.

Table 2: Non-binary benchmarks results (Mean $\pm$ Std.Err. over 25 runs).

| METHOD | BRANIN | PEST CONTROL |
|---|---|---|
| SMAC | 0.6962$\pm$0.0705 | 14.2614$\pm$0.0753 |
| TPE | 0.7578$\pm$0.0844 | 14.9776$\pm$0.0446 |
| SA | 0.4659$\pm$0.0211 | 12.7154$\pm$0.0918 |
| COMBO | **0.4113**$\pm$0.0012 | **12.0012**$\pm$0.0033 |

We exclude BOCS, as the open source implementation provided by the authors does not support ordinal/multi-categorical variables.

**Multi-categorical variables** The Pest control is a modified version of the contamination control with more complex, higher-order variable interactions, as detailed in Supp. 4.2.2. We consider 21 pest control stations, each having 5 choices ($\approx 4.77 \times 10^{14}$ combinatorial choices). We set the budget to 320 including 20 random initial points. Results are in Table 2 and Figure 10 in the Supp. COMBO outperforms all methods and converges faster.

## 4.3 Weighted maximum satisfiability

The satisfiability (SAT) problem is an important combinatorial optimization problem, where one decides how to set variables of a Boolean formula to make the formula true. Many other optimization problems can be reformulated as SAT/MaxSAT problems. Although highly successful, specialized MaxSAT solvers [1] exist, we use MaxSAT as a testbed for BO evaluation. We run tests on three benchmarks from the Maximum Satisfiability Competition 2018.[3] The wMaxSAT weights are unit normalized. All evaluations are negated to obtain a minimization problem. We set the budget to 270 evaluations including 20 random initial points. We report results in Table 3 and Figures in Supp. 4.3, and runtimes on wMaxSAT43 in the figure next to Table 3.on wMaxSAT28 (Figure 14 in the Supp.)[4]

Table 3: (*left*) Negated wMaxSAT Minimum and (*right*) Runtime VS. Minimum on wMaxSAT43.

| Method | wMaxSAT28 | wMaxSAT43 | wMaxSAT60 |
|---|---|---|---|
| SMAC | -20.05$\pm$0.67 | -57.42$\pm$1.76 | -148.60$\pm$1.01 |
| TPE | -25.20$\pm$0.88 | -52.39$\pm$1.99 | -137.21$\pm$2.83 |
| SA | -31.81$\pm$1.19 | -75.76$\pm$2.30 | -187.55$\pm$1.50 |
| BOCS-SDP | -29.49$\pm$0.53 | -51.13$\pm$1.69 | -153.67$\pm$2.01 |
| BOCS-SA3 | -34.79$\pm$0.78 | -61.02$\pm$2.28[a] | N.A[b] |
| COMBO | **-37.80**$\pm$0.27 | **-85.02**$\pm$2.14 | **-195.65**$\pm$0.00 |

[a] 270 evaluations were not finished after 168 hours.
[b] Not tried due to the computation time longer than wMaxSAT43.

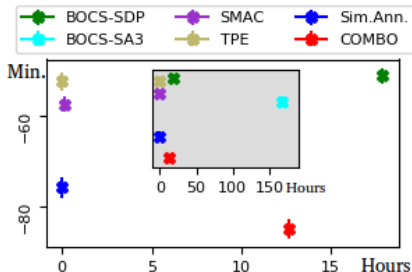

COMBO performs best in all cases. BOCS benefits from third-order interactions on wMaxSAT28 and wMaxSAT43. However, this comes at the cost of large number of parameters [2], incurring expensive computations. When considering higher-order terms BOCS suffers severely from inefficient training. This is due to a bad ratio between the number of parameters and number of training samples (*e.g.*, for the 43 binary variables case BOCS-SA3/SA4/SA5 with, respectively, 3rd/4th/5th order interactions, has 13288/136698/1099296 parameters to train). In contrast, COMBO models arbitrarily high order interactions thanks to GP's nonparametric nature in a statistically efficient way.

Focusing on the largest problem, wMaxSAT60 with $\approx 1.15 \times 10^{18}$ configurations, COMBO maintains superior performance. We attribute this to the sparsity-inducing properties of the Horseshoe prior, after examining non sparsity-inducing priors (Supp.4.3). The Horseshoe prior helps COMBO attain further statistical efficiency. We can interpret this reductionist behavior as the combinatorial version of methods exploiting low-effective dimensionality [3] on continuous search spaces [46].

The runtime –including evaluation time– was measured on a dual 8-core 2.4 GHz (Intel Haswell E5-2630-v3) CPU with 64 GB memory using Python implementations. SA, SMAC and TPE are faster but inaccurate compared to BOCS. COMBO is faster than BOCS-SA3, which needed 168 hours to collect around 200 evaluations. COMBO–modelling arbitrarily high-order interactions– is also faster than BOCS-SDP constrained up-to second-order interactions only.

We conclude that in the realistic maximum satisfiablity problem COMBO yields accurate solutions in reasonable runtimes, easily scaling up to high dimensional combinatorial search problems.

## 4.4 Neural architecture search

Last, we compare BO methods on a neural architecture search (NAS) problem, a typical combinatorial optimization problem [48]. We compare COMBO with BOCS, as well as Regularized Evolution (RE) [38], one of the most successful evolutionary search algorithm for NAS [48]. We include Random Search (RS) which can be competitive in well-designed search spaces [48]. We do not compare with the BO-based NASBOT [23]. NASBOT focuses exclusively on NAS problems and optimizes over a different search space than ours using an optimal transport-based metric between architectures, which is out of the scope for this work.

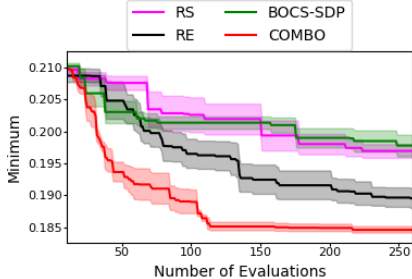

Figure 2: Result for Neural Architecture Search (Mean $\pm$ Std.Err. over 4 runs)

Table 4: (*left*) Connectivity (X – no connection, O – states are connected), (*right)* Computation type.

| | IN | H1 | H2 | H3 | H4 | H5 | OUT |
|---|---|---|---|---|---|---|---|
| IN | - | O | X | X | X | O | X |
| H1 | - | - | X | O | X | X | O |
| H2 | - | - | - | X | O | X | X |
| H3 | - | - | - | - | X | O | X |
| H4 | - | - | - | - | - | O | O |
| H5 | - | - | - | - | - | - | X |
| OUT | - | - | - | - | - | - | - |

| | MAXPOOL | CONV |
|---|---|---|
| SMALL | ID $\equiv$ MAXPOOL($1\times1$) | CONV($3\times3$) |
| LARGE | MAXPOOL($3\times3$) | CONV($5\times5$) |

For the considered NAS problem we aim at finding the optimal cell comprising of one input node (**IN**), one output node (**OUT**) and five possible hidden nodes (**H1**–**H5**). We allow connections from **IN** to all other nodes, from **H1** to all other nodes and so one. We exclude connections that could cause loops. An example of connections within a cell can be found in Table. 4 on the left, where the input state **IN** connects to **H1**, **H1** connects to **H3** and OUT, and so on. The input state and output state have identity computation types, whereas the computation types for the hidden states are determined by combination of 2 binary choices from the table on the right of Table. 4. In total, the search space consists of 31 binary variables, 21 for the connectivities and 2 for 5 computation types.

The objective is to minimize the classification error on validation set of CIFAR10 [26] with a penalty on the amount of FLOPs of a neural network constructed with a given cell. We search for an architecture that balances accuracy and computational efficiency. In each evaluation, we construct a cell, and stack three cells to build a final neural network. More details are given in the Supp. 4.4.

In Figure 2 we can notice that COMBO outperforms other methods significantly. BOCS-SDP and RS exhibit similar performance, confirming that for NAS modeling high-order interactions between variables is crucial. Furthermore, COMBO outperforms the specialized RE, one of the most successful evolutionary search (ES) algorithms shown to perform better on NAS than reinforcement learning (RL) algorithms [38, 48]. When increasing the number of evaluations to 500, RE still cannot reach the performance of COMBO with 260 evaluations, see Figure 17 in the Supp. A possible explanation for such behavior is the high sensitivity to choices of hyperparameters of RE, and ES requires far more evaluations in general. Details about RE hyperparameters can be found in the Supp. 4.4.

Due to the difficulty of using BO on combinatoral structures, BOs have not been widely used for NAS with few exceptions [23]. COMBO's performance suggests that a well-designed general combinatorial BO can be competitive or even better in NAS than ES and RL, especially when computational resources are constrained. Since COMBO is applicable to any set of combinatorial variables, its use in NAS is not restricted to the typical NASNet search space. Interestingly, COMBO can approximately optimize continuous variables by discretization, as shown in the ordinal variable experiment, thus, jointly optimizing the architecture and hyperparameter learning.

## 5  Conclusion

In this work, we propose COMBO, a Bayesian Optimization method for combinatorial search spaces. To the best of our knowledge, COMBO is the first Bayesian Optimization algorithm using Gaussian Processes as a surrogate model suitable for problems with complex high order interactions between variables. To efficiently tackle the exponentially increasing complexity of combinatorial search spaces, we rest upon the following ideas: *(i)* we represent the search space as the combinatorial graph, which combines sub-graphs given to all combinatorial variables using the graph Cartesian product. Moreover, the combinatorial graph reflects a natural metric on categorical choices (Hamming distance) when all combinatorial variables are categorical. *(ii)* we adopt the GFT to define the "smoothness" of functions on combinatorial structures. *(iii)* we propose a flexible ARD diffusion kernel for GPs on the combinatorial graph with a Horseshoe prior on scale parameters, which makes COMBO scale up to high dimensional problems by performing variable selection. All above features together make that COMBO outperforms competitors consistently on a wide range of problems. COMBO is a statistically and computationally scalable Bayesian Optimization tool for combinatorial spaces, which is a field that has not been extensively explored.

## Footnotes

[1]We exclude BOCS from ordinal/multi-categorical experiments, because at the time of the paper submission the open source implementation provided by the authors did not support ordinal/multi-categorical variables. For the explanation on how to use BOCS for ordinal/multi-categorical variables, please refer to the supplementary material of [2].

[2]In [34], the workshop version of this paper, we found that the methods were compared on different sets of initial evaluations and different objectives coming from the random processes involved in the generation of objectives, which turned out to be disadvantageous to COMBO. We reran these experiments making sure that all methods are evaluated on the same set of 25 pairs of an objective and a set of initial evaluations.

[3] https://maxsat-evaluations.github.io/2018/benchmarks.html

[4] The all runtimes were measured on Intel(R) Xeon(R) CPU E5-2630 v3 @ 2.40GHz with python codes.

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
