[Supplementary Material]

# Combinatorial Bayesian Optimization using the Graph Cartesian Product Supplementary Material

## 1 Graph Cartesian product

### 1.1 Graph Cartesian product and Hamming distance

**Theorem 1.1.1.** *Assume a combinatorial graph $\mathcal{G} = (\mathcal{V}, \mathcal{E})$ constructed from categorical variables, $C_1, \cdots, C_N$, that is, $\mathcal{G}$ is a graph Cartesian product $\square_i \mathcal{G}(C_i)$ of complete sub-graphs $\{\mathcal{G}(C_i)\}_i$. Then the shortest path $s(v_1, v_2; \mathcal{G})$ between vertices $v_1 = (c_1^{(1)}, \cdots, c_N^{(1)}), v_2 = (c_1^{(2)}, \cdots, c_N^{(2)}) \in \mathcal{V}$ on $\mathcal{G}$ is equal to the Hamming distance between $(c_1^{(1)}, \cdots, c_N^{(1)})$ and $(c_1^{(2)}, \cdots, c_N^{(2)})$.*

*Proof.* From the graph Cartesian product definition we have that the shortest path between $v_1$ and $v_2$ consists of edges that change a value in one categorical variable at a time. As a result, an edge between $c_i^{(1)}$ and $c_i^{(2)}$, *i.e.*, a difference in the i-*th* categorical variable, and all other edges fixed contributes one error to the Hamming distance. Therefore, we can define the shortest path between $v_1$ and $v_2$ as the sum over all edges for which $c_i^{(1)}$ and $c_i^{(2)}$ are different, $s(v_1, v_2; \mathcal{G}) = \sum_i \mathbb{1}[c_i^{(1)} \neq c_i^{(2)}]$ that is equivalent to the definition of the Hamming distance between two sets of categorical choices. $\square$

### 1.2 Graph Fourier transform with graph Cartesian product

Graph Cartesian products can help us improve the scalability of the eigendecomposition [15]. The Laplacian of the Cartesian product $\mathcal{G}_1 \square \mathcal{G}_2$ of two sub-graphs $\mathcal{G}_1$ and $\mathcal{G}_2$ can be algebraically expressed using the Kronecker product $\otimes$ and the Kronecker sum $\oplus$ [15]:

$$L(\mathcal{G}_1 \square \mathcal{G}_2) = L(\mathcal{G}_1) \oplus L(\mathcal{G}_2) = L(\mathcal{G}_1) \otimes \mathbf{I}_1 + \mathbf{I}_2 \otimes L(\mathcal{G}_2), \tag{6}$$

where $\mathbf{I}$ denotes the identity matrix. As a consequence, considering the eigensystems $\{(\lambda_i^{(1)}, u_i^{(1)})\}$ and $\{(\lambda_j^{(2)}, u_j^{(2)})\}$ of $\mathcal{G}_1$ and $\mathcal{G}_2$, respectively, the eigensystem of $\mathcal{G}_1 \square \mathcal{G}_2$ is $\{(\lambda_i^{(1)} + \lambda_j^{(2)}, u_i^{(1)} \otimes u_j^{(2)})\}$.

**Proposition 1.2.1.** *Assume a graph $\mathcal{G} = (\mathcal{V}, \mathcal{E})$ is the graph cartesian product of sub-graphs $\mathcal{G} = \square_i, \mathcal{G}_i$. Then graph Fourier Transform of $\mathcal{G}$ can be computed in $O(\sum_{i=1}^m |\mathcal{V}_i|^3)$ while the direct computation takes $O(\prod_{i=1}^m |\mathcal{V}_i|^3)$.*

*Proof.* Graph Fourier Transform is eigendecomposition of graph Laplacian $L(\mathcal{G})$ where $\mathcal{G} = (\mathcal{V}, \mathcal{E})$. Eigendecomposition is of cubic complexity with respect to the number of rows(= the number of columns), which is the number of vertices $|\mathcal{V}|$ for graph Laplacian $L(\mathcal{G})$. If we directly compute eigendecomposition of $L(\mathcal{G})$, it costs $O(\prod_i |\mathcal{V}|^3)$. If we utilize graph Cartesian product, then we compute eigendecomposition for each sub-graphs and combine those to obtain eigendecomposition of the original full graph $\mathcal{G}$. The cost for eigendecomposition of each subgraphs is $O(|\mathcal{V}_i|^3)$ and in total, it is summed to $O(\sum_i |\mathcal{V}|^3)$. For graph Cartesian product, graph Fourier Transform can be computed in $O(\sum_i |\mathcal{V}|^3)$. $\square$

**Remark** In the computation of gram matrices, eigenvalues from sub-graphs are summed and entries of eigenvectors are multiplied. Compared to the cost of $O(\prod_i |\mathcal{V}|^3)$, this overhead is marginal. Thus with graph Cartesian product, the ARD diffusion kernel can be computed efficiently with a pre-computed eigensystem for each sub-graphs. This pre-computation is performed efficiently by using Prop. 1.2.1

## 2 Surrogate model fitting

In the surrogate model fitting step of COMBO, GP-parameters are sampled from the posterior using slice sampling [30, 32] as in Spearmint [42, 43].

### 2.1 GP-parameter posterior sampling

For a nonzero mean function, the marginal likelihood of $\mathcal{D} = (V, \mathbf{y})$ is

$$-\frac{1}{2}(\mathbf{y} - m)^T(\sigma_f^2 K_{VV} + \sigma_n^2 I)^{-1}(\mathbf{y} - m) - \frac{1}{2}\log\det(\sigma_f^2 K_{VV} + \sigma_n^2 I) - \frac{n}{2}\log 2\pi \quad (7)$$

where $m$ is the value of constant mean function. With ARD diffusion kernel, the gram matrix is given by

$$\sigma_f^2 K_{VV} + \sigma_n^2 I = \sigma_f^2 \bigotimes_i U_i \exp^{-\beta_i \Lambda_i} U_i^T + \sigma_n^2 I \quad (8)$$

where $\Lambda_i$ is a diagonal matrix whose diagonal entries are eigenvalues of a sub-graph given to a combinatorial variable $L(\mathcal{G}(C_i))$, $U_i$ is a orthogonal matrix whose columns are corresponding eigenvalues of $L(\mathcal{G}(C_i))$, signal variance $\sigma_f^2$ and noise variance $\sigma_n^2$.

**Remark**  In the implementation of eq. (8), a normalized version $\exp^{-\beta_i \Lambda_i} / \Psi_i$ where $\Psi_i = 1/|\mathcal{V}_i| \sum_{j=1,\cdots|\mathcal{V}_i|} \exp^{-\beta_i \lambda_j^{(i)}}$ is used for numerical stability instead of $\exp^{-\beta_i \Lambda_i}$.

In the surrogate model fitting step of COMBO, all GP-parameters are sampled from posterior which is proportional to the product of above marginal likelihood and priors on all GP-parameters such as $\beta_i$'s, signal variance $\sigma_f^2$, noise variance $\sigma_n^2$ and constant mean function value $m$. In COMBO all GP-parameters are sampled using slice sampling [32].

A single step of slice sampling in COMBO consists of multiple univariate slice sampling steps:

1. $m$ : constant mean function value $m$
2. $\sigma_f^2$ : signal variance
3. $\sigma_n^2$ : noise variance
4. $\{\beta_i\}_i$ with a randomly shuffled order

In COMBO, slice sampling does warm-up with 100 burn-in steps and at every new evaluation, 10 more samples are generated to approximate the posterior.

### 2.2 Priors

Especially in BO where data is scarce, priors used in the posterior sampling play an extremely important role. The Horseshoe priors are specified for $\beta_i$'s with the design goal of variable selection as stated in the main text. Here, we provide details about other GP-parameters including constant mean function value $m$, signal variance $\sigma_f^2$ and noise variance $\sigma_n^2$.

#### 2.2.1 Prior on constant mean function value $m$

Given $\mathcal{D} = (V, \mathbf{y})$ the prior over the mean function is the following:

$$p(m) \propto \begin{cases} \mathcal{N}(\mu, \sigma^2) & \text{if} \quad y_{min} \leq m \leq y_{max} \\ 0 & \text{otherwise} \end{cases} \quad (9)$$

where $\mu = mean(\mathbf{y})$, $\sigma = (y_{max} - y_{min})/4$, $y_{min} = \min(\mathbf{y})$ and $y_{max} = \max(\mathbf{y})$.

This is the truncated Gaussian distribution between $y_{min}$ and $y_{max}$ with a mean at the sample mean of $\mathbf{y}$. The truncation bound is set so that untruncated version can sample in truncation bound with the probability of around 0.95.

#### 2.2.2 Prior on signal variance $\sigma_f^2$

Given $\mathcal{D} = (V, \mathbf{y})$ the prior over the log-variance is the following:

$$p(\log(\sigma_f^2)) \propto \begin{cases} \mathcal{N}(\mu, \sigma^2) & \text{if} \quad \frac{\sigma_{\mathbf{y}}^2}{K_{VVmax}} \leq \sigma_f^2 \leq \frac{\sigma_{\mathbf{y}}^2}{K_{VVmin}} \\ 0 & \text{otherwise} \end{cases} \quad (10)$$

where $\sigma_{\mathbf{y}}^2 = variance(\mathbf{y})$, $\mu = \frac{1}{2}(\frac{\sigma_{\mathbf{y}}^2}{K_{VVmin}} + \frac{\sigma_{\mathbf{y}}^2}{K_{VVmax}})$, $\sigma = \frac{1}{4}(\frac{\sigma_{\mathbf{y}}^2}{K_{VVmin}} + \frac{\sigma_{\mathbf{y}}^2}{K_{VVmax}})$, $K_{VVmin} = \min(K_{VV})$, $K_{VVmax} = \max(K_{VV})$ and $K_{VV} = K(V, V)$.

This is the truncated Log-Normal distribution. The intuition behind this choice of prior is that in GP prior, $\sigma_f^2 K_{VV}$ is covariance matrix of $\mathbf{y}$ with the assumption of very small noise variance $\sigma_n^2$. The truncation bound is set so that untruncated version can sample in truncation bound with the probability of around 0.95. Since for larger $\sigma_f^2$, the the magnitude of the change of $\sigma_f^2$ has less significant effect than for smaller $\sigma_f^2$. In order to take into account relative amount of change in $\sigma_f^2$, the Log-Normal distribution is used rather than the Normal distribution.

### 2.2.3 Priors on scaling factor $\beta_i$ and noise variance $\sigma_n^2$

We use the Horseshoe prior for $\beta_i$ and $\sigma_n^2$ in order to encourage sparsity. Since the probability density of the Horseshoe is intractable, its closed form bound is used as a proxy [7]:

$$\frac{K}{2} \log(1 + \frac{4\tau^2}{x^2}) < p(x) < K \log(1 + \frac{2\tau^2}{x^2}) \tag{11}$$

where $x = \beta_i$ or $x = \sigma_n^2$, $\tau$ is a global shrinkage parameter and $K = (2\pi^3)^{-1/2}$ [7]. Typically, the upper bound is used to approximate Horseshoe density. For $\beta_i$, we use $\tau = 5$ to avoid excessive sparsity. For $\sigma_n^2$, we use $\tau = \sqrt{0.05}$ that prefers very small noise similarly to the Spearmint implementation.[5]

### 2.3 Slice Sampling

At every new evaluation in COMBO, we draw samples of $\beta_i$. For each $\beta_i$ the sampling procedure is the following:

SS-1 Set $t = 0$ and choose a starting $\beta_i^{(t)}$ for which the probability is non-zero.

SS-2 Sample a value $q$ uniformly from $[0, p(\beta_i^{(t)}|\mathcal{D}, \beta_{-i}^{(t)}, m^{(t)}, (\sigma_f^2)^{(t)}, (\sigma_n^2))^{(t)}]$.

SS-3 Draw a sample $\beta_i$ uniformly from regions, $p(\beta_i|\mathcal{D}, \beta_{-i}^{(t)}, m^{(t)}, (\sigma_f^2)^{(t)}, (\sigma_n^2)^{(t)}) > q$.

SS-4 Set $\beta_i^{(t+1)} = \beta_i$ and repeat from SS-2 using $\beta_i^{(t+1)}$.

In SS-2, we step out using doubling and shrink to draw a new value. For detailed explanation about slice sampling, please refer to [32]. For other GP-parameters, the same univariate slice sampling is applied.

## 3 Acquisition function maximization

In the acquisition function maximization step, we begin with candidate vertices chosen to balance between exploration and exploitation. $20,000$ vertices are randomly selected for exploration. To balance exploitation, we use 20 spray vertices similar to spray points[6] in [42]. Spray vertices are randomly chosen in the neighborhood of a vertex with the best evaluation (e.g, $nbd(v_{best}, 2) = \{v|d(v, v_{best}) \leq 2\}$). Out of $20,020$ initial vertices, 20 vertices with the highest acquisition values are used as initial points for further optimization. This type of combination of heuristics for exploration and exploitation has shown improved performances [13, 28].

We use a breadth-first local search (BFLS) to further optimize the acquisition function. At a given vertex we compare acquisition values on adjacent vertices. We then move to the vertex with the highest acquisition value and repeat until no acquisition value on an adjacent vertex is higher than acquisition value at the current vertex.

`spearmint/spearmint/chooser/GPEIOptChooser.py#L235`

## 3.1 Non-local search for acquisition function optimization

We tried simulated annealing as a non-local search in different ways, namely:

1. Randomly split 20 initial points into 2 sets of 10 points and optimize from 10 points in one set with BFLS and optimize from 10 points in another set with simulated annealing.

2. For given 20 initial points, optimize from 20 points with BFLS and optimize from the same 20 points with simulated annealing.

3. For given 20 initial points, firstly optimize from 20 points with BFLS and use 20 points optimized by BFLS as initial points for optimization using simulated annealing.

The optimum of all 3 methods is hardly better than the optimum discovered solely by BFLS. Therefore, we decided to stick to the simpler procedure without SA.

# 4 Experiments

## 4.1 Bayesian optimization with binary variables

### 4.1.1 Ising sparsification

Ising sparsification is about approximating a zero-field Ising model expressed by $p(z) = \frac{1}{Z_p} \exp\{z^\top J^p z\}$, where $z \in \{-1, 1\}^n$, $J^p \in \mathbb{R}^{n \times n}$ is an interaction symmetric matrix, and $Z_p = \sum_z \exp\{z^\top J^p z\}$ is the partition function, using a model $q(z)$ with $J_{ij}^q = x_{ij} J_{ij}^p$ where $x_{ij} \in \{0, 1\}$ are the decision variables. The objective function is the regularized Kullback-Leibler divergence between $p$ and $q$.

$$\mathcal{L}(x) = D_{KL}(p\|q) + \lambda\|x\|_1 \tag{12}$$

where $\lambda > 0$ is the regularization coefficient $D_{KL}$ could be calculated analytically [2]. We follow the same setup as presented in [2], namely, we consider $4 \times 4$ grid of spins, and interactions are sampled randomly from a uniform distribution over $[0.05, 5]$. The exhaustive search requires enumerating all $2^{24}$ configurations of $x$ that is infeasible. We consider $\lambda \in \{0, 10^{-4}, 10^{-2}\}$. We set the budget to 170 evaluations.

| Method | $\lambda = 0.0$ |
|---|---|
| SMAC | 0.1516±0.0404 |
| TPE | 0.4039±0.1087 |
| SA | **0.0953**±0.0331 |
| BOCS − SDP | 0.1049±0.0308 |
| COMBO | 0.1030±0.0351 |

Figure 3: Ising sparsification ($\lambda = 0.0$)

Figure 4: Ising sparsification ($\lambda = 0.0001$)

| Method | $\lambda = 0.0001$ |
|---|---|
| SMAC | 0.2192±0.0522 |
| TPE | 0.4437±0.0952 |
| SA | 0.1166±0.0353 |
| BOCS − SDP | **0.0586**±0.0125 |
| COMBO | 0.0812±0.0279 |

Figure 5: Ising sparsification ($\lambda = 0.01$)

| Method | $\lambda = 0.01$ |
|---|---|
| SMAC | 0.3501±0.0447 |
| TPE | 0.6091±0.1071 |
| SA | 0.3342±0.0636 |
| BOCS − SDP | **0.3001**±0.0389 |
| COMBO | 0.3166±0.0420 |

#### 4.1.2 Contamination control

The contamination control in food supply chain is a binary optimization problem [19]. The problem is about minimizing the contamination of food where at each stage a prevention effort can be made to decrease a possible contamination. Applying the prevention effort results in an additional cost $c_i$. However, if the food chain is contaminated at stage $i$, the contamination spreads at rate $\alpha_i$. The contamination at the $i$-th stage is represented by a random variable $\Gamma_i$. A random variable $z_i$ denotes a fraction of contaminated food at the $i$-th stage, and it could be expressed in an recursive manner, namely, $z_i = \alpha_i(1 - x_i)(1 - z_{i-1}) + (1 - \Gamma_i x_i)z_{i-1}$, where $x_i \in \{0, 1\}$ is the decision variable representing the preventing effort at stage $i$. Hence, the optimization problem is to make a decision at each stage whether the prevention effort should be applied so that to minimize the general cost while also ensuring that the upper limit of contamination is $u_i$ with probability at least $1 - \varepsilon$. The initial contamination and other random variables follow beta distributions that results in the following objective function

$$\mathcal{L}(x) = \sum_{i=1}^{d}\left[c_i x_i + \frac{\rho}{T}\sum_{k=1}^{T}1_{\{z_k > u_i\}}\right] + \lambda\|x\|_1 \tag{13}$$

where $\lambda$ is a regularization coefficient, $\rho$ is a penalty coefficient (we use $\rho = 1$) and we set $T = 100$. Following [2], we assume $u_i = 0.1$, $\varepsilon = 0.05$, and $\lambda \in \{0, 10^{-4}, 10^{-2}\}$. We set the budget to 270 evaluations.

Figure 6: Contamination control ($\lambda = 0.0$)

| Method | $\lambda = 0.0$ |
|---|---|
| SMAC | 21.4644±0.0312 |
| TPE | 21.6408±0.0437 |
| SA | 21.4704±0.0366 |
| BOCS − SDP | 21.3748±0.0246 |
| COMBO | **21.2752**±0.0292 |

Figure 7: Contamination control ($\lambda = 0.0001$)

| Method | $\lambda = 0.0001$ |
|---|---|
| SMAC | 21.5011±0.0329 |
| TPE | 21.6868±0.0406 |
| SA | 21.4871±0.0372 |
| BOCS − SDP | 21.3792±0.0296 |
| COMBO | **21.2784**±0.0314 |

Figure 8: Contamination control ($\lambda = 0.01$)

| Method | $\lambda = 0.01$ |
|---|---|
| SMAC | 21.6512±0.0403 |
| TPE | 21.8440±0.0422 |
| SA | 21.6120±0.0385 |
| BOCS − SDP | 21.5232±0.0269 |
| COMBO | **21.4436**±0.0293 |

## 4.2 Bayesian optimization with ordinal and multi-categorical variables

### 4.2.1 Oridinal variables : discretized branin

In order to test COMBO on ordinal variables. We adopt widely used continuous benchmark branin function. Branin is defined on $[0, 1]^2$, we discretize each dimension with 51 equally space points so

that center point can be chosen in the discretized space. Therefore, the search space is comprised of 2 ordinal variables with 51 values(choices) for each.

COMBO outperforms all of its competitors. In Figure 9, SMAC and TPE exhibit similar search progress as COMBO, but in term of the final value at 100 evaluations, those two are overtaken by SA. COMBO maintains its better performance over all range of evaluations up to 100.

| Method | Branin |
|---|---|
| SMAC | 0.6962±0.0705 |
| TPE | 0.7578±0.0844 |
| SA | 0.4659±0.0211 |
| COMBO | **0.4113**±0.0012 |

Figure 9: Ordinal variables : discretized branin

### 4.2.2 Multi-categorical variables : pest control

| Method | Pest |
|---|---|
| SMAC | 14.2614±0.0753 |
| TPE | 14.9776±0.0446 |
| SA | 12.7154±0.0918 |
| COMBO | **12.0012**±0.0033 |

Figure 10: Multi-categorical variables : pest control

In the chain of stations, pest is spread in one direction, at each pest control station, the pest control officer can choose to use a pesticide from 4 different companies which differ in their price and effectiveness.

For $N$ pest control stations, the search space for this problem is $5^N$, 4 choices of a pesticide and the choice of not using any of it.

The price and effectiveness reflect following dynamics.

- If you have purchased a pesticide a lot, then in your next purchase of the same pesticide, you will get discounted proportional to the amount you have purchased.
- If you have used a pesticide a lot, then pests will acquire strong tolerance to that specific product, which decrease effectiveness of that pesticide.

Formally, there are four variables: at $i$-th pest control $Z_i$ is the portion of the product having pest, $A_i$ is the action taken, $C_i^{(l)}$ is the adjusted cost of pesticide of type $l$, $T_i^{(l)}$ is the beta parameter of the

Beta distribution for the effectiveness of pesticide of type $l$. It starts with initial $Z_0$ and follows the same evolution as in the contamination control, but after each choice of pesticide type whenever the taken action is to use one out of 4 pesticides or no action. $\{C_i^{(l)}\}_{1,\cdots,4}$ are adjusted in the manner that the pesticide which has been purchased most often will get a discount for the price. $\{T_i^{(l)}\}_{1,\cdots,4}$ are adjusted in the fashion that the pesticide which has been frequently used in previous control point cannot be as effective as before since the insects have developed tolerance to that.

The portion of the product having pest follows the dynamics below

$$z_i = \alpha_i(1 - x_i)(1 - z_{i-1}) + (1 - \Gamma_i x_i)z_{i-1} \tag{14}$$

when the pesticide is used, the effectiveness $x_i$ of pesticide follows beta distribution with the parameters, which has been adjusted according to the sequence of actions taken in previous control points.

Under this setting, our goal is to minimize the expense for pesticide control and the portion of products having pest while going through the chain of pest control stations. The objective is similar to the contamination control problem

$$\mathcal{L}(x) = \sum_{i=1}^{d} \left[ c_i x_i + \frac{\rho}{T} \sum_{k=1}^{T} 1_{\{z_k > u_i\}} \right] \tag{15}$$

However, we want to stress out that the dynamics of this problem is far more complex than the one in the contamination control case. First, it has 25 variables and each variable has 5 categories. More importantly, the price and effectiveness of pesticides are dynamically adjusted depending on the previously made choice.

### 4.3 Weighted maximum satisfiability(wMaxSAT)

Satisfiability problem is the one of the most important and general form of combinatorial optimization problems. SAT solver competition is held in Satisfiability conference every year.[7] Due to the resemblance between combinatorial optimizations and weighted Maximum satisfiability problems, in which the goal is to find boolean values that maximize the combined weights of satisfied clauses, we optimize certain benchmarks taken from Maximum atisfiability(MaxSAT) Competition 2018. We took randomly 3 benchmarks of weighted maximum satisfiability problems with no hard clause with the number of variables not exceeding 100.[8] The weights are normalized by mean subtraction and standard deviation division and evaluations are negated to be minimization problems.

| Method | 28 |
|---|---|
| SMAC | -20.0530±0.6735 |
| TPE | -25.2010±0.8750 |
| SA | -31.8060±1.1929 |
| BOCS-SDP | -29.4865±0.5348 |
| BOCS-SA3 | -34.7915±0.7814 |
| COMBO | **-37.7960**±0.2662 |

Figure 11: wMaxSAT28

Figure 12: wMaxSAT43*BOCS-SA3 was run for 168 hours but could not finish 270 evaulations.

| Method | 43 |
|---|---|
| SMAC | -57.4217±1.7614 |
| TPE | -52.3856±1.9898 |
| SA | -75.7582±2.3048 |
| BOCS-SDP | -51.1265±1.6903 |
| BOCS-SA3* | -61.0186±2.2812 |
| COMBO | **-85.0155**±2.1390 |

Figure 13: wMaxSAT60

| Method | 60 |
|---|---|
| SMAC | -148.6020±1.0135 |
| TPE | -137.2104±2.8296 |
| SA | -187.5506±1.4962 |
| BOCS-SDP | -153.6722±2.0096 |
| COMBO/GM | -152.0745±8.5167 |
| COMBO | **-195.6527**±0.0000 |

Figure 14: Runtime VS. Minimum on wMaxSAT28

Figure 15: Runtime VS. Minimum on wMaxSAT4. BOCS-SA3 did not finish all 270 evaluations after 168 hours, we plot the runtime for BOCS-SA3 as 168 hours.

## 4.4 Neural architecture search(NAS)

### 4.4.1 Search space

Table 5: (*left*) Connectivity and (*right)* Computation type.

|      | IN | H1 | H2 | H3 | H4 | H5 | OUT |
|------|----|----|----|----|----|----|-----|
| IN   | -  | O  | X  | X  | X  | O  | X   |
| H1   | -  | -  | X  | O  | X  | X  | O   |
| H2   | -  | -  | -  | X  | O  | X  | X   |
| H3   | -  | -  | -  | -  | X  | O  | X   |
| H4   | -  | -  | -  | -  | -  | O  | O   |
| H5   | -  | -  | -  | -  | -  | -  | X   |
| OUT  | -  | -  | -  | -  | -  | -  | -   |

|       | MAXPOOL | CONV |
|-------|---------|------|
| SMALL | ID $\equiv$ MAXPOOL($1\times1$) | CONV($3\times3$) |
| LARGE | MAXPOOL($3\times3$) | CONV($5\times5$) |

In our architecture search problem, the cell consists of one input state(**IN**), one output state(**OUT**) and five hidden states(**H1**∼**H5**). The connectivity between 7 states are specified as in the left of Table. 5 where it can be read that (**IN**→**H1**) and (**IN**→**H5**) from the first row. Input and output states are identity maps. The computation type of each of 5 hidden states are determined by combination of 2 binary choices as in the right of Table. 5.

In total, our search space consists of 31 binary variables.[9]

### 4.4.2 Evaluation

For a given 31 binary choices, we construct a cell and stack 3 cells as follows

<div align="center">

Input
↓
Conv($3 \times 16 \times 3 \times 3$)-BN-ReLU
↓
Cell with 16 channels
↓
MaxPool($2 \times 2$)-Conv($16 \times 32 \times 1 \times 1$)
↓
Cell with 32 channels

</div>

$$\downarrow$$
$$\text{MaxPool}(2 \times 2)\text{-Conv}(32 \times 64 \times 1 \times 1)$$
$$\downarrow$$
$$\text{Cell with 64 channels}$$
$$\downarrow$$
$$\text{MaxPool}(2 \times 2)\text{-FC}(1024 \times 10)$$
$$\downarrow$$
$$\text{Output}$$

At each MaxPool, the height and the width of features are halved.

The network is trained for 20 epochs with Adam [24] with the default settings in pytorch [36] except for the weight decay of $5 \times 10^{-5}$. CIFAR10 [26] training data is randomly shuffled with random seed 0 in the command "numpy.RandomState(0).shuffle(indices)". In the shuffled training data, the first 30000 is used for training and the last 10000 is used for evaluations. Batch size is 100. Early stopping is applied when validation accuracy is worse than the validation accuracy 10 epochs ago.

Due to the small number of epochs, evaluations are a bit noisy. In order to stabilize evaluations, we run 4 times of training for a given architecture. On GTX 1080 Ti with 11GB, 4 runs can be run in parallel. Depending on a given architecture training took approximately 5∼30 minutes.

Since the some binary choices result in invalid architectures, in such case, validation accuracy is given as 10%, which is the expected accuracy of constant prediction.

The final evaluation is given as

$$Error_{validation} + 0.02 \cdot \frac{\text{FLOPs of a given architecture}}{\text{Maximim FLOPs in the search space}} \qquad (16)$$

where "Maximim FLOPs in the search space" is computed from the cell with all connectivity among states and Conv($5 \times 5$) for all **H1**∼**H5**. 0.02 is set with the assumption that we can afford 1% of increased error with 50% reduction in FLOPs.

| Method | NAS |
|---|---|
| RS | 0.1969±0.0011 |
| BOCS − SDP | 0.1978±0.0017 |
| RE | 0.1895±0.0016 |
| COMBO | **0.1846**±0.0005 |

Figure 16: Neural architecture search experiment.

### 4.4.3 Comparison to NASNet search space

| Binary | | NASNet |
|---|---|---|
| Yes | Invalid Architecture | No |
| Not fixed | The number of inputs to each state | 2 |
| 4 | The number of computation type of states | 13 |

### 4.4.4 Regularized evolution hyperparameters

In evolutionary search algorithms, the choice of mutation is critical to the performance. Since our binary search space is different from NASNet search space where Regulairzed Evolution(RE)

Figure 17: Neural architecture search experiment with additional evaluations for RE (up to 500 evaluations).

| Method(#eval) | NAS |
|---|---|
| RE(260) | 0.1895±0.0016 |
| RE(500) | 0.1888±0.0019 |
| COMBO(260) | **0.1846**±0.0005 |

was originally applied, we modify mutation steps. All possible mutations proposed in [38] can be represented as simple binary flipping in binary search space. In binary search space, some binary variables are about computation type and others are about connectivity. Thus uniform-randomly choosing binary variable to flip can mutate computation type or connectivity. Since binary search space is redundant we did not explicitly include identity mutation (not mutating). Since evolutionary search algorithms are believed to be less sample efficient than BO, we gave an advantage to RE by only allowing valid architectures in mutation steps.

On other crucial hyperparameters, population size $P$ and sample size $S$, motivated by the best value used in [38], $P = 100$, $S = 25$. We set our $P$ and $S$ to have similar ratio as the one originally proposed. Since, we assumed less number of evaluations(260, 500) compared to 20000 in [38], we reduced $P$ and $S$. In NAS on binary search space, we used $P = 50$ and $S = 15$.

## Footnotes

[5]https://github.com/JasperSnoek/spearmint

[6]`https://github.com/JasperSnoek/spearmint/blob/b37a541be1ea035f82c7c82bbd93f5b4320e7d91/`

[7]http://satisfiability.org/, http://sat2018.azurewebsites.net/competitions/

[8]https://maxsat-evaluations.github.io/2018/benchmarks.html maxcut-johnson8-2-4.clq.wcnf (28 variables), maxcut-hamming8-2.clq.wcnf (43 variables), frb-frb10-6-4.wcnf (60 variables)

[9]We design a binary search space for NAS so that to also compare with BOCS. COMBO is not restricted to binary choices for NAS, however.