[Reviews · NeurIPS 2019]

Reviewer 1



Beautiful approach to deal with categorical and integer variables in Bayesian Optimization that builds a graph with the combinations of these variables. Necessary approach to continue the research in making BO able to deal these variables. When I read papers such as this one I am willing for it to be accepted. I do not have much to say rather than good job. And I say this being an author that have proposed another approach in the same scenario. If we add the quality of the paper with the sensational supplementary material, the theoretical material added and the quality of the experiments, I heavily recommend for acceptance this paper. It is a paper that I will give a seminar to my colleagues without a doubt and maybe, who knows, expand it. Congratulations to the authors. The code would have been nice to be uploaded. I just wish to say to the authors that I would like to see an extension of this paper published in a journal, it would be very great.

Reviewer 2



This manuscript proposes a system for combinatorial Bayesian optimization called COMBO, aimed at problems with large numbers of categorical and/or ordinal features. The main contribution is an effective kernel for this setting based on applying a graph kernel to the graph Cartesian product of each of the features, which can be computed efficiently by exploiting structure. This kernel can be further enhanced using an ARD extension and a horseshoe prior to encourage sparse feature selection. The COMBO system then creates a GP with this kernel and does random + local search to maximize an acquisition function such as EI in the combinatorial space. A series of experiments demonstrate COMBO performing better on real and synthetic tasks than alternatives such as systems using one-hot encodings. Overall this is a well-written paper and was interesting to read. I think the proposed kernel may find use in Bayesian optimization or elsewhere, and would be a nice addition to a GP/BayesOpt software packages. There would certainly be interest in this contribution by attendees of the conference. That said I do have some concerns, mostly minor, that I will enumerate below in mostly linear (rather than importance) order: - I wish the manuscript could be more self-contained. There are a so many forward references to the supplementary material that it can be somewhat frustrating for the reader. I understand the page limits are sometimes oppressive, but for some of these I wish they were at least complemented by some intuition or summary to keep the flow. For example the use of a horseshoe prior is mentioned several times as a main contribution and responsible for a large increase in performance, but this is not really justified in the paper except a pointer to the supplement in line 277. It would be nice to say something about this result beyond "The Horseshoe prior helps COMBO..." Added after response: I wish you had addressed the above comment in your response. I continue to feel the manuscript as written is not as effective as it could be. - I wanted to point to a couple more papers using Bayesian optimization in (different) combinatorial spaces: - Garnett, et al. Bayesian Optimization for Sensor Set Selection. IPSN 2010. [optimization over subsets of a ground set] - Malkomes, et al. Bayesian optimization for automated model selection. NeurIPS 2016. [optimization over a graph defining GP kernels] - I personally think stating and setting off Theorem 2.2.1 as a theorem is overkill, as the result is in my opinion both intuitive and straightforward. It needlessly breaks the flow of the text. - I feel the same way about Proposition 2.3.1. I think the paper would be nicer to read if this were simply stated as a normal sentence. (Again I find this result unsurprising, albeit of course convenient!) - [*] Perhaps the biggest technical concern I had was about the optimization of the acquisition function in COMBO, which is of course also a combinatorial optimization problem. The authors simply do random search followed by local refinement. Although this will help identify good "exploration moves," it is not clear to me that this will help with "exploitation moves," when they are warranted, as it's not clear that you'd be able to stumble onto the part of the domain near the best-seen points using this method, where the optimum will lie in the latter case. Note that the two papers mentioned above deal with this problem by combining an exploration heuristic (effectively the same, random selection) with an exploitation heuristic (measuring the acquisition function at points near the best-seen observations). The fact that was not done here makes me wonder whether the improved performance of COMBO may simply be a result of biasing EI toward more exploration than it would normally do. The current experiments give no insight into this possibility. Added after response: thank you for your clarification. - Are the standard errors in Table 2 correct? The standard deviation for COMBO was only 0.006? Did it find the same point every time but once or something? Added after response: I still find it extremely suspicious that the standard error would be so absurdly small with no real explanation given. Yes there is no observation noise, but there are other sources of randomness that seem like they should contribute non-negligable variance. [getting very minor now] - Table 3 would be nicer to read if the -'s were typeset as negative signs ($-20.05$) rather than hyphens (-20.05) - The use of dashes in the manuscript --en dashes not set off by spaces on the side of the parenthetical comment-- is so idiosyncratic and nonstandard to the point of being jarring. - Please read over the references, there are some typos (e.g., gaussian).

Reviewer 3



The authors describe a novel method for Bayesian optimization in discrete input spaces. The method is based on using combinatorial graphs and their graph Laplacing to specify a covariance function for a Gaussian process. An efficient implementation is described. Quality: The proposed method is of high quality: well-motivated, with significant theory and validated through an exhaustive series of experiments. Clarity: The paper is very well written and easy to read. Originality: The proposed method is novel up to my knowledge. This is the first time that a method based on combinatorial graphs is applied to the problem of Bayesian optimization with discrete inputs. Significance: The contribution is highly significant. The proposed method is shown to be clearly better than existing baselines in exhaustive experiments. The approach respresents a new way of solving Bayesian optimization problems in discrete spaces.

[Author Response · NeurIPS 2019]

We thank all the reviewers for their insightful and encouraging feedback.

**Initial points of acquisition function optimization**  Following the implementation details of Spearmint [1], COMBO adopts the idea of *spray points* [2] to promote exploitation, which is the same method as in the reference pointed by **R2**. Due to the discrete nature of COMBO's search space, the implementation detail is slightly different. In Spearmint, spray points are points around the best-evaluation point, after perturbing by a zero-mean Gaussian with user-specified variance (e.g $0.001^2$). In contrast, in COMBO's combinatorial graphs, we have spray vertices. Spray vertices are randomly chosen vertices neighboring the best evaluated vertex, such that the shortest path distance to the best vertex is less than or equal to a user-specified distance (e.g, 2). As **R2** suggested, this heuristic promotes exploitation.

Using random vertices for exploration is similar to Spearmint. In Spearmint we use Sobol sequences, whereas in COMBO random vertices are chosen uniformly. For initial points for the acquisition function optimization, we probe acquisition values on 20,020 points consisted of 20 spray vertices and 20,000 random vertices, similar to Spearmint. The best 20 from 20,020 points are used as initial points for further optimization. In the camera-ready version we will give the full array of details regarding the exploitation-exploration trade-off, including the references pointed by **R2**.

**Computational complexity**  In addition to the surrogate model fitting and acquisition function optimization in ordinary Bayesian optimization methods, COMBO has a one-off pre-processing step of eigendecomposition to compute Fourier basis. The complexity of the one-off eigendecomposition is *linear* with respect to the number of variables.

*Surrogate model fitting.* 1-dimensional slice sampling is used with the Gibbs sampler. The entire sampling procedure has $O(k^2 d^2) + O(kd^3)$ complexity. Specifically, the 1-dimensional slice sampling requires *(a)* $O(kd^2)$ multiplications for $k$ variables and $d$ evaluations and *(b)* solving a linear system of cubic complexity $O(d^3)$. The $O(kd^2)$ multiplications, specifically, are due to the Kronecker product kernel that the diffusion kernel yields on a Graph Cartesian product.

*Acquisition function optimization.* We observe that the heuristic of local optimization with initial points selected from the pool of spray vertices and random vertices (similar to the description given by **R2**) works better than more complex solutions with simulated annealing as stated in the paper. We leave the trade-off between optimization and efficiency of other methods, such as, evolutionary search and genetic algorithm as future work.

**Error measure**  The error measure in the experiment is the standard error. The evaluations in all experiments except NAS are noise-free. Thus, small yet still different from 0 standard errors indicate different optima.

**Future works and limitations.**  COMBO currently handles continuous variables only if they are discretized. Mixing combinatorial and continues variables is challenging for Bayesian Optimization, not only for designing an appropriate surrogate model but also for the optimization of the acquisition function.

Besides addressing the aforementioned limitations, bridging the well-established graph theory with combinatorial optimization opens up lots of interesting research directions (**R1**). Examples are irregular structures not amenable to the graph Cartesian product or even learning the graph structure. Further, exploring problems, like joint neural architecture and hyperparameter search, traditionally dominated by low sample efficiency methods (*e.g.,* evolutionary search) is of great interest: the superior uncertainty quantification of the Bayesian Optimization surrogate model, like Gaussian Processes, would bring great advantages.

As pointed out by reviewers, we hope the graph theory framework on combinatorial optimization introduced by COMBO will attract attention to the under-explored problem of combinatorial Bayesian optimization. We appreciate the meticulous comments from all reviewers on the structure, presentation and typos. We will include all the clarifications in the camera-ready version and release all code, experiments and results upon acceptance.

## Footnotes

[1]J. Snoek, H. Larochelle, R.P. Adams. Practical Bayesian Optimization of Machine Learning Algorithms. NeurIPS 2012. (https://github.com/JasperSnoek/spearmint)

[2]https://github.com/JasperSnoek/spearmint/blob/b37a541be1ea035f82c7c82bbd93f5b4320e7d91/spearmint/spearmint/chooser/GPEIOptChooser.py#L235


[Meta-Review · NeurIPS 2019]

Two reviewers really liked the idea of the paper. R2 remained concerned about the validity of the experiments, given the very small standard errors. Please address all reviewer concerns in the camera ready version.